

# Genome-wide identification and characterization of the NF-Y proteins in *Zanthoxylum armatum*

Xianzhe Zheng, Yanling Duan, Huifang Zheng, Hao Tang, Liumeng Zheng and Xiaobo Yu

Southwest Research Center for Cross Breeding of Special Economic Plants, School of Life Science, Leshan Normal University, Leshan, China

## ABSTRACT

Somatic embryogenesis from nucellar tissues is common in many *Zanthoxylum* plants. The nuclear factor Y (NF-Y) transcription factor is pivotal in this process. Despite its significance, the identification and functional analysis of the NF-Y transcription factor family in *Zanthoxylum armatum* (*Z. armatum*) remains unexplored. This study identified 67 *ZaNF-Y* transcription factors in the diploid *Z. armatum* genome, comprising 20 *ZaNF-YA*, 30 *ZaNF-YB*, and 17 *ZaNF-YC* genes. Gene duplication, conserved domain, and motif analyses revealed the similarity and specificity of *ZaNF-Y* members in functional evolution. Cis-element analysis suggested that plant hormones and various transcription factor families may regulate *ZaNF-Y* gene expression, impacting nucellar embryo formation. Expression analysis across tissues indicated that the expression of most *ZaNF-Y* genes, such as *ZaNF-YB5*, was low in female flowers. In contrast, *ZaNF-YC1* was highly expressed in female flowers and young fruit, indicating their potential role in nucellar embryo formation. Additionally, protein association network analysis provided insights into the composition of *ZaNF-Y* complexes. Our study enhances understanding of *ZaNF-Y* transcription factors and provides a basis for harnessing apomixis in hybrid crop production.

## INTRODUCTION

Apomixis is a form of asexual reproduction that occurs without fertilization stage, which is divided into sporophytic apomixis and gametophytic apomixis (*Wang & Underwood, 2023*). Dandelion is a typical plant that undergoes gametophytic apomixis, where the diploid egg cell forms an embryo through parthenogenesis (*Underwood et al., 2022*). Existing studies have shown that PARTHENOGENESIS (*PAR*) is a key regulatory gene for the apomixis in dandelion (*Underwood et al., 2022*). The ectopic expression of BABY BOOM1 (*BBM1*) gene in rice egg cell can also induce the formation of somatic embryos without fertilization (*Khanday et al., 2019*). In sporophytic apomixis, the embryo originates from somatic cells within the ovule, such as nucellus cells. *Citrus* is a typical plant exhibiting sporophytic apomixis. Previous studies have shown that the RWP-RK transcription factor *CitRWP* plays a crucial regulatory role in the process of apomixis in *citrus* (*Wang*

Corresponding author
Xiaobo Yu, yuxiaobosky@163.com

*et al., 2017*). *Zanthoxylum armatum* DC., an aromatic plant of the *Rutaceae* family, is economically important for its applications in traditional Chinese medicine and cuisine (*Hui et al., 2020*). Several species in the *Zanthoxylum* genus, including *Z. armatum*, exhibit sporophytic apomixis, where somatic embryos develop from nucellar cells, a process similar to that in *Citrus* plants (*Hojsgaard & Pullaiah, 2022*). Studies on apomixis in *Zanthoxylum* have identified genes such as *Zardc07021*, which is homologous to the *CitRWP* gene in *Citrus*, known for its role in apomixis (*Wang et al., 2017*). Heterologous expression in *Arabidopsis* affects floral organ development (*Wang et al., 2021*). Overexpression of the D-class MADS-box transcription factor, *ZbAGL11*, from *Zanthoxylum bungeanum*, enables seed formation in normally emasculated *Arabidopsis* lines (*Fei et al., 2021*). Recently, preliminary analyses explored the function of the RWP-RK transcription factor family in apomixis of *Z. armatum* (*Zheng et al., 2024*). However, understanding the transcriptional regulatory mechanisms of somatic embryo formation from the nucellus in *Z. armatum* remains limited.

Nuclear factor Y (NF-Y), also called the CCAAT-box binding factor (CBF) or heme-associated protein, is a family of transcription factors conserved across eukaryotes (*Mantovani, 1999*; *Myers & Holt 3rd, 2018*). The family consists of three subunits: NF-YA, NF-YB, and NF-YC (*Nardini et al., 2013*). Typically, NF-YB and NF-YC form a heterodimer in the cytoplasm, which then binds to NF-YA in the nucleus to assemble a functional NF-Y heterotrimer complex (*Gnesutta et al., 2017*). This complex regulates downstream gene expression either through the NF-YA subunit's N-terminal or by interacting with other regulatory proteins (*Nardone, Chaves-Sanjuan & Nardini, 2017*; *Huang et al., 2015*). NF-Y transcription factors play vital roles in plant growth, development, stress responses, and hormone signaling (*Kavi Kishor et al., 2023*; *Zhang et al., 2023*). For instance, overexpression of *AtNF-YA5* enhanced the drought tolerance by reducing leaf water loss in *Arabidopsis* (*Li et al., 2008*). Similarly, overexpression of the homologous gene *GmNF-YA5* in soybeans resulted in the same phenotype as *AtNF-YA5* (*Ma et al., 2020*). Under long-day conditions, the overexpression of *OsHAP5H* (*OsNF-YC9*) delays heading in rice (*Li et al., 2016*). In soybean, *GmNF-YC9* is a homologous gene to *OsNF-YC9* and is involved in the regulation of drought and salt stress resistance, but it does not affect flowering time (*Yu et al., 2024*). This indicates that the functions of the *NF-Y* genes are both conserved and unique across different plants.

In recent decades, NF-Y family members have been extensively studied for their roles in embryo development. LEAFY COTYLEDON 1 (*LEC1*, *AtNF-YB9*), a key transcription factor, is essential for embryo maturation and exhibits a distinct expression pattern in embryonic tissues (*Lotan et al., 1998*; *Lee et al., 2003*). Overexpression of *LEC1* in transgenic plants induces somatic embryo-like structures (*Harada, 2001*). Similarly, in rice, the *LEC1* homolog *OsNF-YB7* is predominantly expressed in embryos, and its disruption results in abnormal embryo formation (*Niu et al., 2021*). Other related proteins, like LEC1-like (*LIL*, *NF-YB6*), also play critical roles in embryo development (*Kwong et al., 2003*). In Arabidopsis, *NF-YA3* and *NF-YA8* are functionally redundant but collectively necessary for embryogenesis (*Fornari et al., 2013*). *GhL1L1* regulates somatic embryogenesis in cotton by influencing auxin distribution (*Xu et al., 2019*). However, in *Z. armatum*, the

identification and functional analysis of NF-Y transcription factors in nucellar embryo formation are still poorly understood. A clear understanding of expression patterns or CRISPR-based knockouts is vital for improving our knowledge of NF-Y function.

In this study, we identified 67 *ZaNF-Y* genes in *Z. armatum*, comprising 20 *ZaNF-YA*, 30 *ZaNF-YB*, and 17 *ZaNF-YC* genes. An in-depth analysis was conducted on these genes, including chromosomal localization, physicochemical properties, subfamily classification, conserved domains, and cis-elements. We also examined the potential functions of *ZaNF-Y* genes in nucellar embryo formation through tissue expression variations and protein interaction predictions. Our study provides a comprehensive understanding of *ZaNF-Y* transcription factors in *Z. armatum* and suggests promising directions for future research on the regulatory mechanisms of nucellar embryo formation.

## MATERIALS & METHODS

### Identification and basic characteristics of *ZaNF-Y* family members

Using the reported diploid *Z. armatum* genome data (https://doi.org/10.6084/m9.figshare. 14400884.v1) and model profiles of NF-YA (PF02045) and NF-YB/C (PF00808) from the Pfam database (http://pfam-legacy.xfam.org/), we predicted the ZaNF-Y proteins with HMMER software (version 3.0), applying a cut-off value of 0.01 based on previous studies (*Liu et al., 2021*; *Wang et al., 2021*; *Prakash et al., 2017*). All ZaNF-Y proteins were screened using the National Center for Biotechnology Information Conserved Domain Database (https://www.ncbi.nlm.nih.gov/cdd/). We determined the location, strand, and coding sequences (CDS) of *ZaNF-Y* genes based on the *Z. armatum* genome data. The CDS of *ZaNF-Y* genes are shown in File S1. ProtParam is a tool that enables the calculation of physical and chemical properties for a protein sequence provided by the user. The physicochemical properties of ZaNF-Y proteins, including the number of amino acids, molecular weights, and pI, were obtained using ProtParam tools (http://web.expasy.org/protparam/). Subcellular localization of ZaNF-Y proteins was predicted using the WoLF PSORT tool (https://wolfpsort.hgc.jp/), which was trained on protein sequences of more than 14,000 organisms and predicted based on amino acid sequences, with an overall prediction accuracy of over 80%.

### Phylogenetic analysis of *ZaNF-Y* family members

AtNF-Y protein sequences from *Arabidopsis* were downloaded from the *Arabidopsis* Information Resource website (http://www.arabidopsis.org/) as previously reported (*Siefers et al., 2009*). OsNF-Y protein sequences from rice were obtained from the MSU TIGR database (http://rice.uga.edu/) as previously reported (*Yang et al., 2017*). CgNF-Y protein sequences from *Citrus grandis* were sourced from the Citrus Pan-genome2breeding database (http://citrus.hzau.edu.cn/index.php/) as previously reported (*Mai et al., 2019*). The amino acid sequences of NF-Y proteins from *Z. armatum*, *Arabidopsis*, rice and *Citrus grandis* are shown in File S2. The Multiple Alignment using the Fast Fourier Transform program was used to align NF-Y proteins from *Z. armatum*, *Arabidopsis*, *Oryza sativa*, and *C. grandis* (*Katoh, Rozewicki & Yamada, 2019*). RaxmlGUI serves as a graphical user interface for RAxML, a highly popular software for phylogenetic inference utilizing

maximum likelihood (*Edler et al., 2021*). Phylogenetic trees were constructed using the maximum likelihood method with 1,000 bootstrap replications in the raxmlGUI program (version 2.0) and only nodes with bootstrap values above 70% were considered reliable. These trees were annotated using EvolView (https://www.evolgenius.info/evolview-v2/).

## Chromosomal location and gene duplication

The chromosomal locations of *ZaNF-Y* genes were determined and visualized using TBtools-II (*Chen et al., 2023*). MCScanX is one of the commonly used tools for detecting gene colinearity and evolutionary analysis (*Wang et al., 2012*). Segmental duplications of *ZaNF-Y* members were identified through the multiple collinear scanning toolkits (MCScanX) using parameters such as a match score of 50, a gap penalty of −1, a match size of 5, an e value of 1e−05, and a maximum gap count of 25 (*Wang et al., 2012*). The resulting collinearity file was visualized using TBtools-II (*Chen et al., 2023*). Homologous genes between *Z. armatum* and other species (*Arabidopsis*, *O. sativa*, and *C. grandis*) were identified, and synteny analysis was conducted using the MCscanX tool and visualized using TBtools-II (*Chen et al., 2023*; *Wang et al., 2012*).

## Analysis of conserved structural domains and conserved motifs of ZaNF-Y proteins

Conserved structural domains of ZaNF-Y proteins were identified using CD-search (https://www.ncbi.nlm.nih.gov/Structure/bwrpsb/bwrpsb.cgi) (*Wang et al., 2023*). Conserved motifs in ZaNF-Y proteins were analyzed with MEME (https://meme-suite.org/meme/tools/meme) with the following parameters: A maximum of 10 motifs and motif widths ranging from six to 50. The results were visualized using TBtools-II (*Chen et al., 2023*).

## Cis-element and binding transcription factors prediction of *ZaNF-Y* promoters

The promoters of *ZaNF-Y* genes (regions upstream of the start codon within two kb) were extracted from the *Z. armatum* genome data using TBtools-II software. PlantCare (Plant Cis-acting Regulatory Element) is a database focusing on plant cis-regulatory elements (such as promoters, enhancers, repressors, *etc.*), but its updates are lagging behind (*Lescot et al., 2002*). Plant Promoter Analysis Navigator (PlantPAN) is mainly used to predict transcription factor binding sites (TFBS) and binding transcription factors, with a wide range of species coverage and timely data updates (*Chow et al., 2024*). The promoter sequences of *ZaNF-Y* genes were submitted to PlantCARE (https://bioinformatics.psb.ugent.be/webtools/plantcare/html/) and Plant Promoter Analysis Navigator (PlantPAN) (version 4.0) (http://plantpan.itps.ncku.edu.tw/plantpan4/promoter_analysis.php) for cis-element and binding transcription factors prediction. The results were visualized using TBtools-II (*Chen et al., 2023*). The promoter sequences of *ZaNF-Y* genes are shown in File S3. The cis-acting elements of *ZaNF-Y* genes are shown in File S4. The transcription factors that may be bound to the *ZaNF-Ys'* promoter are shown in File S5.

## RNA extraction and reverse transcription

To verify the expression of *ZaNF-Y* genes in different tissues, we collected female flowers, young fruits, stems, and leaves of *Z. armatum*, following a previous study (*Zheng et al.,*
*2024*). Total RNA was extracted using the FastPure® Plant Total RNA Isolation Kit (Polysaccharides and Polyphenolics-rich) (Vazyme, Nanjing, China), and cDNA was prepared using the ABScript III RT Master Mix for qPCR with gDNA Remover (ABclonal, RK20428).

### qRT-PCR analysis

qRT-PCR was performed using a 2X Universal SYBR Green Fast qPCR Mix (ABclonal, RK21203). The qRT-PCR reaction protocol consisted of one cycle at 95 °C for 30 s, followed by 40 cycles at 95 °C for 10 s and 60 °C for 30 s, with a melting curve generated automatically. The *ZaGAPDH* gene was used as an internal control according to our previous study (unpublished results), and the description are detailed in File S6. The relative expression levels of *ZaNF-Y* genes were calculated using the $2-\Delta\Delta Ct$ method. Statistical analyses were conducted using the International Business Machines Statistical Package for the Social Sciences software (version 27.0). The primers used in qRT-PCR are detailed in File S7. The raw data for Ct values are shown in File S8. The MIQE checklist is shown in File S6.

### Protein association network analysis of ZaNF-Y proteins

STRING serves as an online database frequently employed to develop protein-protein interaction networks, scoring each interaction between target proteins (*Szklarczyk et al., 2023*). The protein association network of ZaNF-Y proteins was analyzed using STRING (version 12.0; https://cn.string-db.org/).

### Statistical analysis

The qRT-PCR experiment included three biological and three technical replicates to ensure the reproducibility. Data were analyzed using one-way ANOVA with Tukey's tests. Significant differences were defined as those with a *P* value less than 0.05.

## RESULTS

### Genome-wide identification and physicochemical properties of *ZaNF-Y* Genes

We identified 67 *ZaNF-Y* transcription factors in the *Z. armatum* genome using NF-YAs (PF02045) and NFYBs/Cs (PF00808) from the Pfam database. To classify these factors, we constructed a phylogenetic tree using amino acid sequences of 36 *Arabidopsis* AtNF-Y proteins (*Siefers et al., 2009*), 34 rice OsNF-Y proteins (*Yang et al., 2017*), and 24 pummelo CgNF-Y proteins (*Mai et al., 2019*) based on sequence similarity and conservation of NF-Y proteins among different species (Fig. 1). The ZaNF-Y proteins clustered into three subfamilies: NF-YA, NF-YB, and NF-YC. The NF-YA subfamily contains 21 members (ZaNF-YA1–ZaNF-YA20), the NF-YB subfamily contains 30 members (ZaNF-YB1–ZaNF-YB30), and the NF-YC subfamily contains 17 members (ZaNF-YC1–ZaNF-YC17) (Fig. 1, File S9). Based on chromosomal locations, 10 *ZaNF-Y* genes (*ZaNF-YA18-20* and *ZaNF-YB24-30*) were not annotated to specific chromosomes (Fig. 2). Analysis of their physical and chemical properties revealed coding sequences ranging from 285 bp (*ZaNF-YC3*) to 1,020 bp (*ZaNF-YA1*), corresponding to amino acid sequences from 94 aa (ZaNF-YC3)
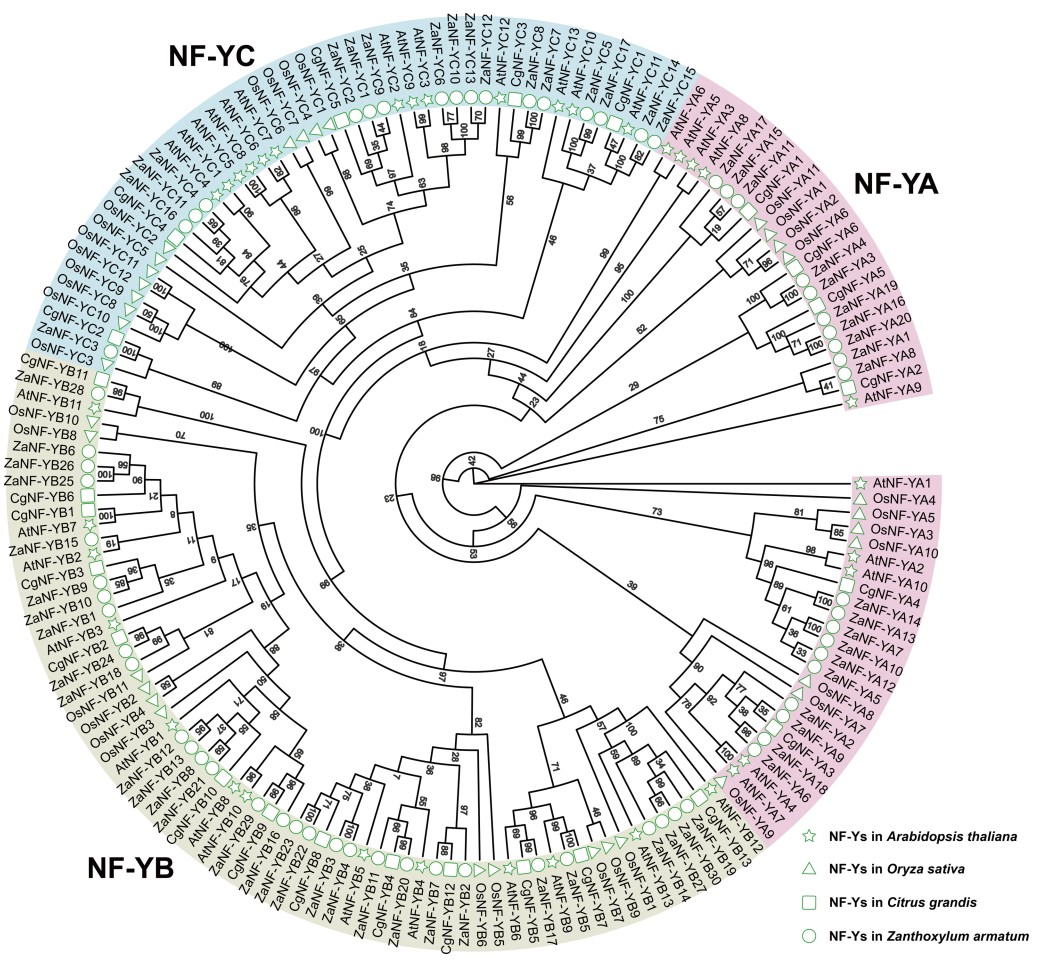

**Figure 1** **Phylogenetic relationships between ZaNF-Y proteins and other plant NF-Y proteins.** Raxml-GUI 2.0 was used to construct the maximum likelihood (ML) tree with 1,000 bootstrap replicates and displayed using EvolView.

to 339 aa (ZaNF-YA1), with molecular weights between 10.35 kDa (ZaNF-YC3) and 37.13 kDa (ZaNF-YA15) (File S9). The isoelectric points (pI) values ranged from 4.53 (ZaNF-YB14) to 9.89 (ZaNF-YA14) (File S9). Most *ZaNF-Y* members were predicted to localize in the nucleus (59), with a few in the cytoplasm (8) (File S9).

## Gene duplication of *ZaNF-Y* genes

Gene duplications are crucial for gene expansion and evolution. Analyzing gene family expansions and duplications benefits significantly from collinearity information. For *ZaNF-Y* genes, 16 pairs of segmental duplications and two tandemly duplicated genes (*ZaNF-YC12* and *ZaNF-YC13*) were identified (Fig. 3). Collinearity often indicates homologous sequences with potentially similar functions. We constructed three collinearity maps to examine collinearity between *ZaNF-Y* and *NF-Y* genes in *Arabidopsis*, *O. sativa*, and *C. grandis* (Fig. 4). The results identified 33, six, and 39 pairs of orthologous genes between *Z. armatum* and *Arabidopsis*, *O. sativa*, and *C. grandis*, respectively. Given the extensive
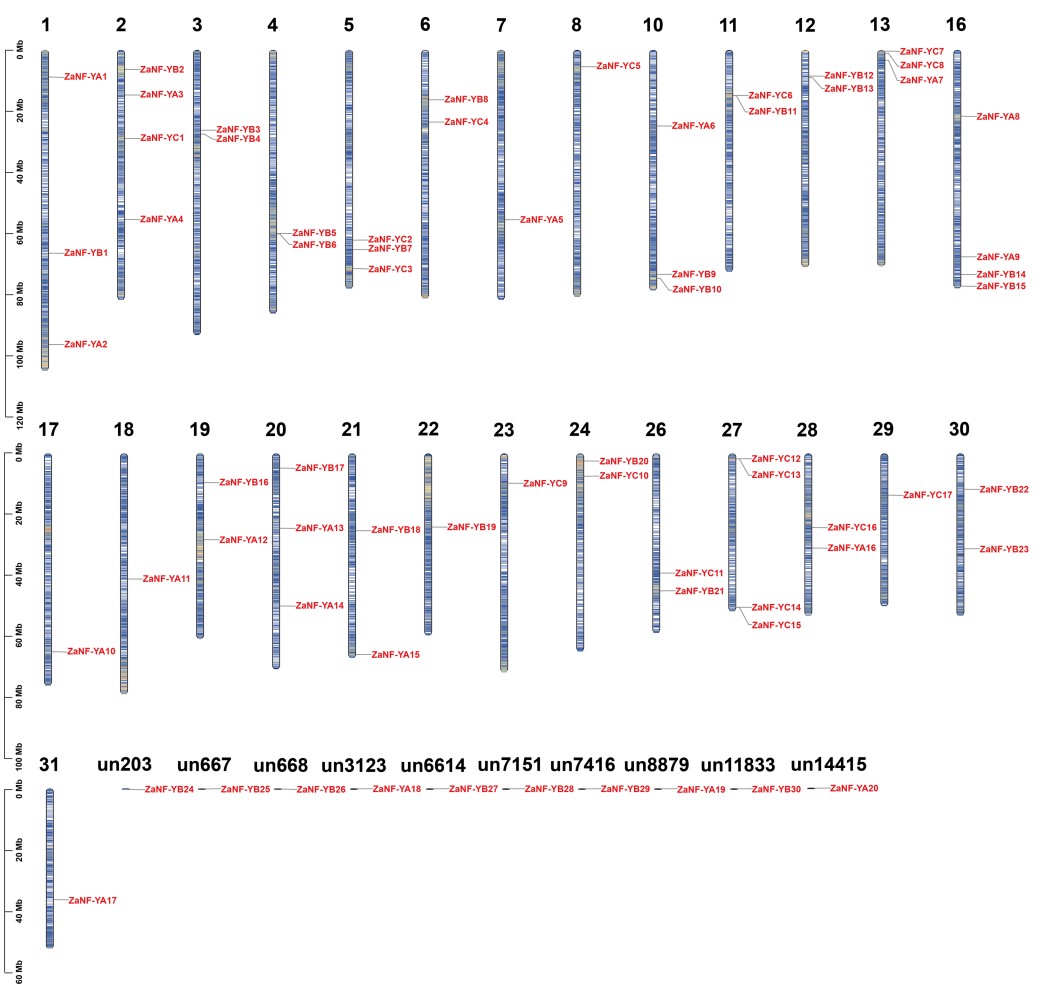

**Figure 2  Chromosomal locations of the *ZaNF-Y* genes.** The vertical bars represent the chromosomes with numbers at the top of each bar representing chromosome number. "un" represents the unassembled scaffold. *ZaNF-Y* genes are numbered in the order of chromosomes.

research on *NF-Y* genes in other plants, identifying these homologous pairs can enhance our understanding of *ZaNF-Y* gene functions in *Z. armatum*.

## Conserved structural domains and motifs analysis of ZaNF-Y proteins

Phylogeny, conserved domains, motif information, and domain conservation were used to assess relationships among *ZaNF-Y* family members (Fig. 5). Using the MEME software, we identified 10 conserved motifs in ZaNF-Y proteins, ranging from 15 to 50 amino acids (Figs. 5B and 5D). The ZaNF-YB subfamily mainly contains Motifs 1, 2, and 5. Motifs 7 and 3 are predominant in the ZaNF-YA subfamily, while Motifs 4, 6, and 10 are common in the ZaNF-YC subfamily. Additionally, Motif 9 is unique to certain ZaNF-YA members (ZaNF-YA5, ZaNF-YA7, ZaNF-YA10, ZaNF-YA12, and ZaNF-YA13), suggesting it may play a crucial role in these genes. Conserved structural domains of ZaNF-Y proteins were also identified (Fig. 5C). The CBF domain characterizes the main NF-YA subfamily, while

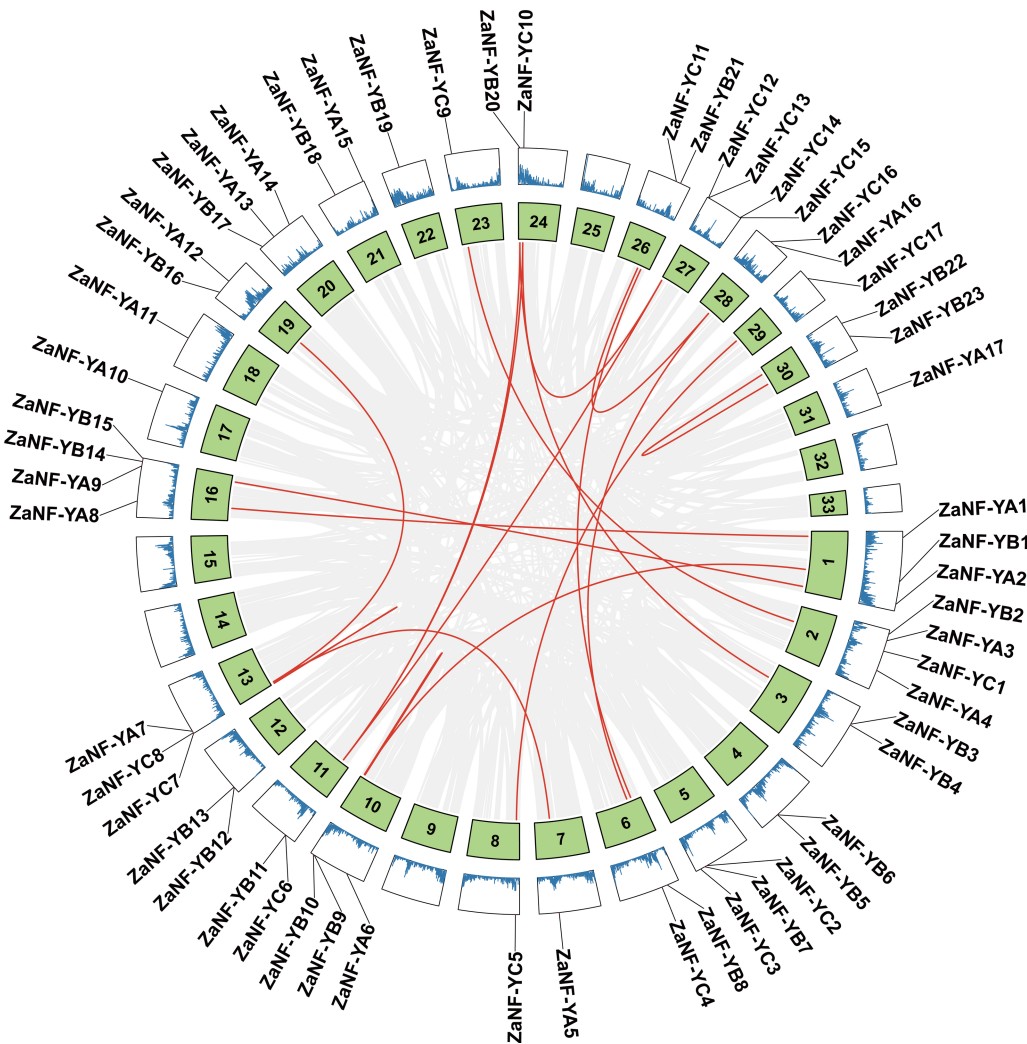

**Figure 3** **Duplication analysis of the *ZaNF-Y* genes on different chromosomes.** The red lines represent collinearity relationship of *ZaNF-Y* genes. The gray lines connect all genes with a collinearity relationship in *Z. armatum*. Chromosome numbers are indicated by the texts in the green boxes. The blue lines in the outermost box indicate gene density. *ZaNF-Y* genes are indicated on the chromosomes with a black line.

the HFD-NFYB domain is typical for most ZaNF-YB members. Most NF-YC subfamily members contain the HFD_NFYC-like domain. Some proteins possess unique domains; for example, ZaNF-YB28 has the HFD_POLE3_DPB4 domain, and the HFD_DRAP1 domain is exclusive to ZaNF-YC14 and ZaNF-YC15. These findings suggest that these proteins may have significant regulatory roles through their unique HFD domains.

## Cis-element analysis of *ZaNF-Y* promoters

The PlantCare database is commonly used for analyzing cis-elements associated with factors like hormones and abiotic stresses in promoter. In order to determine whether *ZaNF-Y* genes are regulated by these factors during the formation of nucellar embryo, cis-elements in the promoter regions 2,000 bp upstream of the start codon were identified

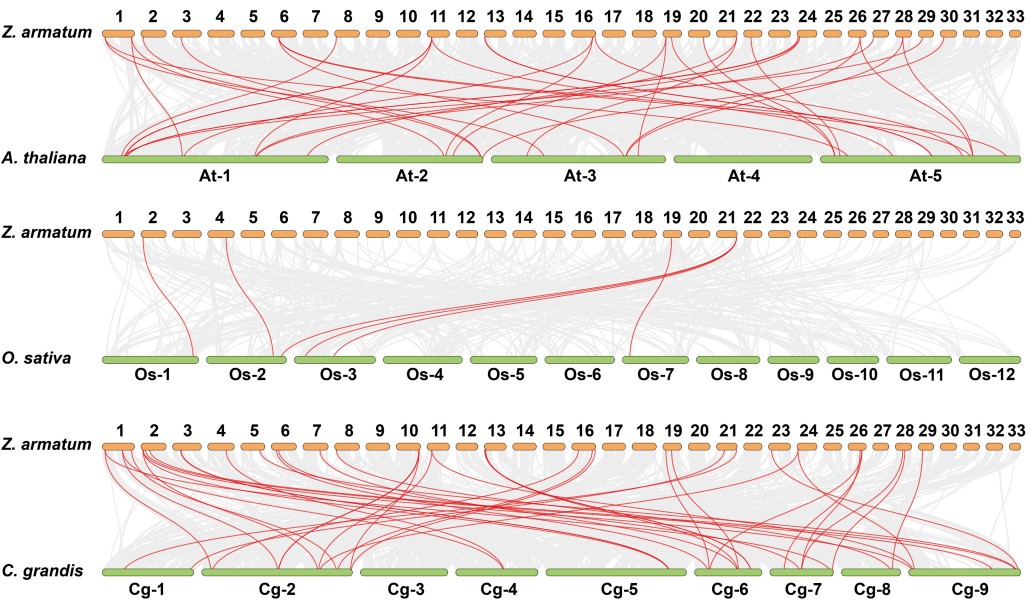

**Figure 4 Synteny analysis of _ZaNF-Y_ genes between _Z. armatum, Arabidopsis thaliana, Oryza sativa_, and _Citrus grandis_.** Za, _Z. armatum_; At, _Arabidopsis thaliana_; Os, _Oryza sativa_; Cg, _Citrus grandis_. The red lines indicate the collinearity relationship between NF-Y genes. Colinear gene pairs between _Z. armatum_ and three other species are indicated by the gray lines in the backdrop.

using PlantCare (Fig. 6, File S4). A total of 21 types of regulatory elements were identified in the promoter regions of 67 _ZaNF-Y_ genes, classified into three categories as previous research: plant growth and development, phytohormone response, and abiotic stress (_Hu et al., 2022_). Most elements were phytohormone-responsive, including 170 ABA response elements, 34 auxin-responsive elements (23 TGA-element and 11 AuxRR-core), and 65 gibberellin-responsive elements (29 P-box, 20 GARE-motif, and 16 TATC-box). Plant growth and development-related elements formed the second largest category. Certain cis-elements were specific to particular _ZaNF-Y_ genes, such as HD-Zip 3 (_ZaNF-YA13_) and Box III (_ZaNF-YB1_ and _ZaNF-YB29_). In addition, the PlantPan database is a useful tool for identifying transcription factors in promoter region. The results showed that the promoter regions of _ZaNF-Y_ genes contained multiple transcription factor family members, such as AP2, AT-hook, bZIP, bHLH, and GATA (Fig. 7, File S5). These transcription factors may regulate _ZaNF-Y_ gene expression, affecting physiological processes in _Z. armatum_, including nucellar embryo formation.

## Tissue-specific expression of _ZaNF-Y_ genes

In the apomixis process of _Zanthoxylum_ plants, female flowers can develop into complete fruits without pollination (_Fei et al., 2021_). During this process, nucellar cells develop into somatic embryos, which ultimately form seeds. To investigate the potential function of _ZaNF-Y_ genes in nucellar embryo formation in _Z. armatum_, we measured their expression levels in female flowers and young fruits using quantitative RT-PCR (qRT-PCR) (Fig. 8). Expression levels in stems and leaves served as controls. Most _ZaNF-Y_ genes exhibited
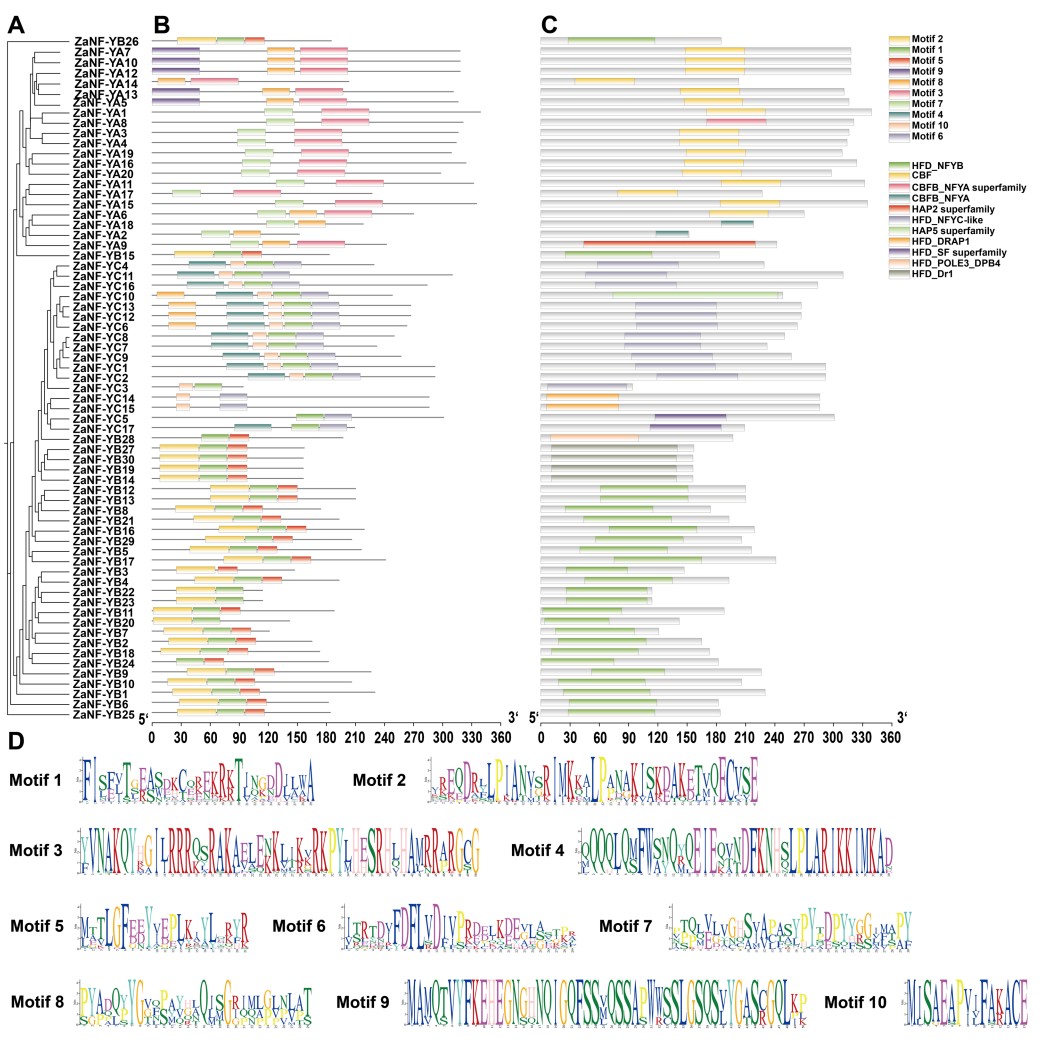

**Figure 5** **Phylogenetic relationships, conserved motifs, and conserved structural domains of ZaNF-Y proteins.** (A) Phylogenetic tree of 67 ZaNF-Y proteins. (B) Distribution of conserved motifs in ZaNF-Y proteins. Colors distinguish each of the ten motifs. (C) Distribution of conserved domains in ZaNF-Y proteins. Each of the eleven domains is distinguished by different colors. (D) Sequences logos of the 10 conserved motifs in ZaNF-Y proteins.

low expression in female flowers, suggesting their involvement in regulating somatic embryogenesis. Specifically, *ZaNF-YA1*, *ZaNF-YA3*, *ZaNF-YA15*, *ZaNF-YB5*, *ZaNF-YB6*, *ZaNF-YB7*, *ZaNF-YB9*, and *ZaNF-YC12* were significantly downregulated in female flowers and young fruits. The expressions of *ZaNF-YB5*, *ZaNF-YB6*, and *ZaNF-YC12* were notably lower in female flowers and young fruits compared to stems and leaves. In contrast, *ZaNF-YA9*, *ZaNF-YA16*, *ZaNF-YA18*, and *ZaNF-YC6* were highly expressed in young fruits, while *ZaNF-YC1* was highly expressed in both female flowers and young fruits, indicating their potential roles in apomixis.

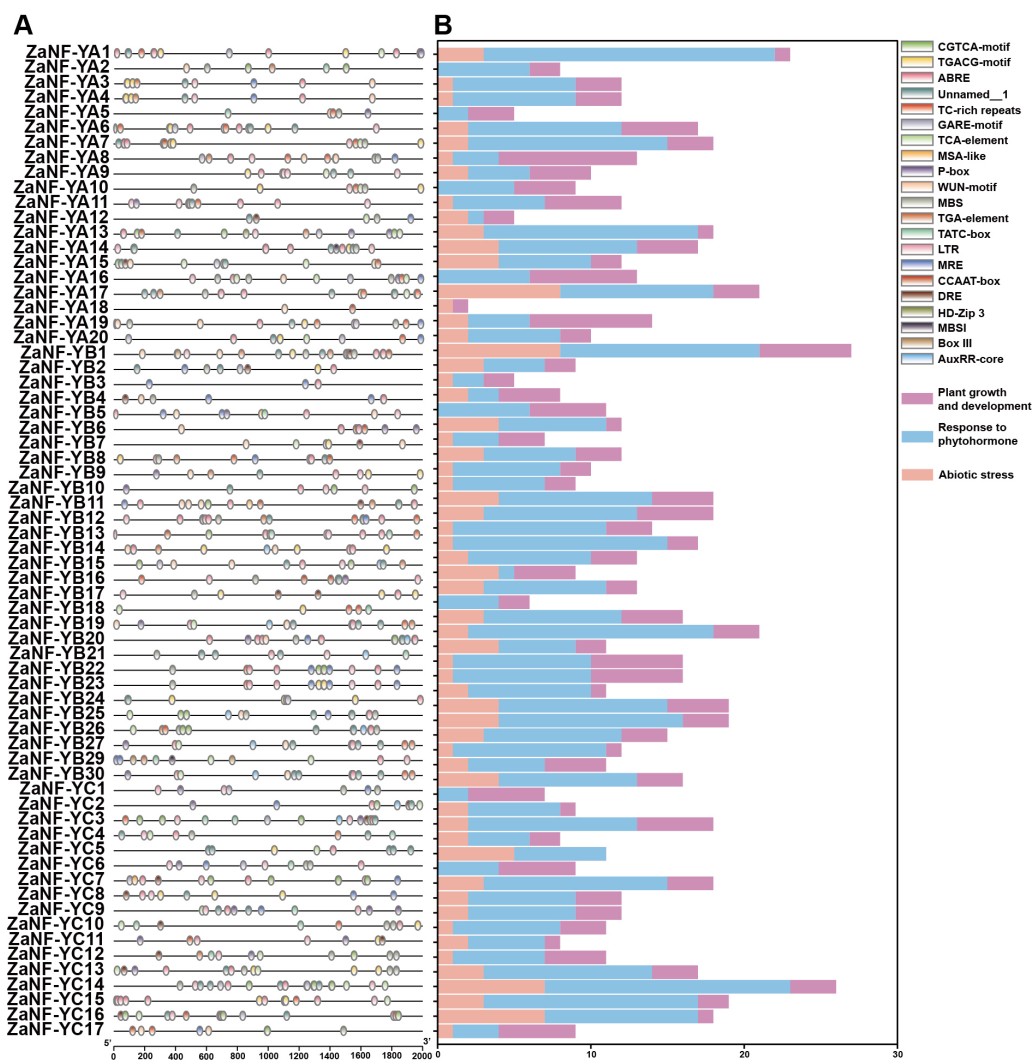

**Figure 6  Cis-element analysis of the promoter of the *ZaNF-Y* genes.** (A) Location of cis-elements in the *ZaNF-Y* promoter regions. Each of the cis-elements is distinguished by different colors. (B) Statistical analysis of cis-elements for each *ZaNF-Y* gene in three categories.

## Correlation analysis of apomixis-related *ZaNF-Y* genes

Previous research has demonstrated that ectopic expression of the *LEC1* gene can induce somatic embryo-like structures (*Harada, 2001*). Phylogenetic tree analysis depicts that *ZaNF-YB5* in *Z. armatum* is closely related to *AtLEC1* (Fig. 1). Typically, NF-YB and NF-YC proteins form heterodimers, which then associate with NF-YA proteins to form heterotrimers. Using the STRING database, we analyzed the potential interacting proteins of ZaNF-YB5 (Fig. 9). The results indicated that ZaNF-YC3, ZaNF-YC6, ZaNF-YC7, ZaNF-YC8, ZaNF-YC13, and ZaNF-YC16 might interact with ZaNF-YB5. Additionally, ZaNF-YB5 could interact with ZaNF-YA1, ZaNF-YA3, ZaNF-YA4, ZaNF-YA9, ZaNF-YA11, ZaNF-YA12, ZaNF-YA15, and ZaNF-YA19, suggesting that these ZaNF-Y proteins might form heterotrimers with regulatory roles. Considering the high expression of

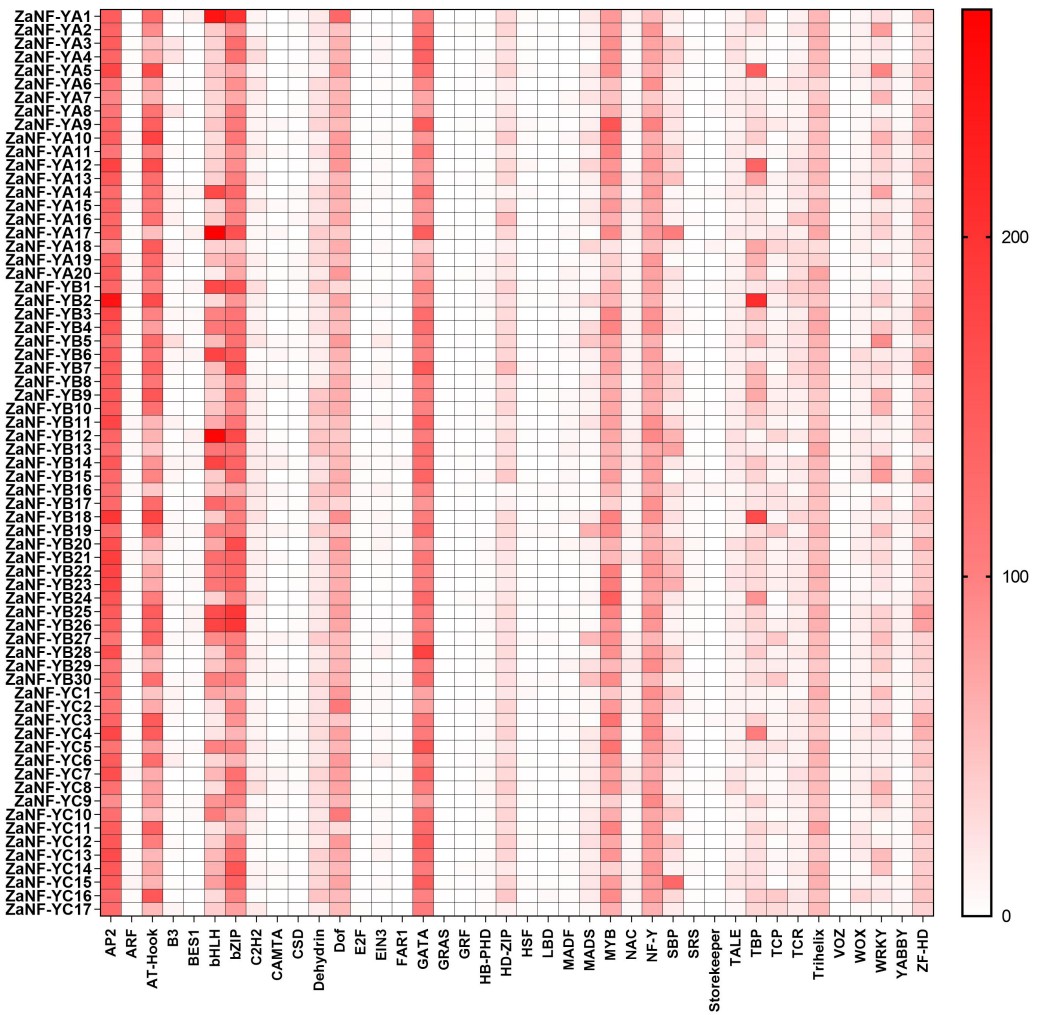

**Figure 7 Statistical analysis of transcription factor binding sites in the *ZaNF-Y* promoter regions.**
Binding sites are more abundant in colors with darker tints.

ZaNF-YC1 in female flowers and young fruits, we also predicted its interacting proteins. The results indicated that members of the ZaNF-YA subfamily (ZaNF-YA9, ZaNF-YA11, ZaNF-YA12, and ZaNF-YA15) and members of the ZaNF-YB subfamily (ZaNF-YB6, ZaNF-YB8, ZaNF-YB12, ZaNF-YB13, ZaNF-YB16, ZaNF-YB17, ZaNF-YB24, ZaNF-YB28, and ZaNF-YB29) might interact with ZaNF-YC1. Some ZaNF-YA subfamily members (ZaNF-YA9, ZaNF-YA11, ZaNF-YA12, and ZaNF-YA15) were predicted to interact with both ZaNF-YB5 and ZaNF-YC1, suggesting these proteins might form heterotrimers with key regulatory roles in nucellar embryo formation.

## DISCUSSION

NF-Y transcription factors play vital roles in biological processes such as flowering, stress response, growth, and development, as shown in species like cabbage, melon,

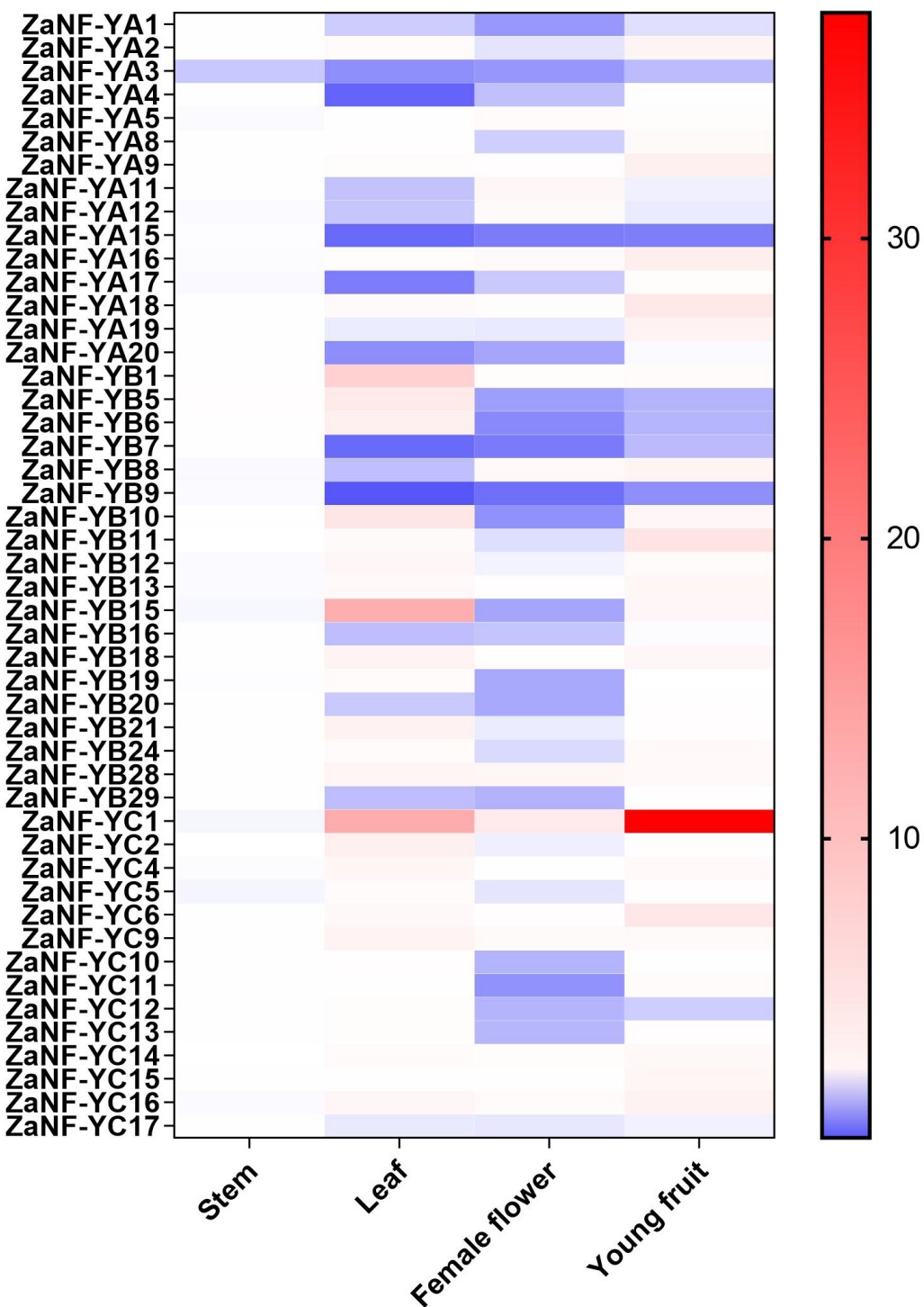

**Figure 8 Expression patterns analysis of *Za NF-Y* genes in different tissues (stem, leaf, female flowers, and young fruit) based on the qRT-PCR data.** Different colours represent different expression levels. The expression levels of stem were used as control.

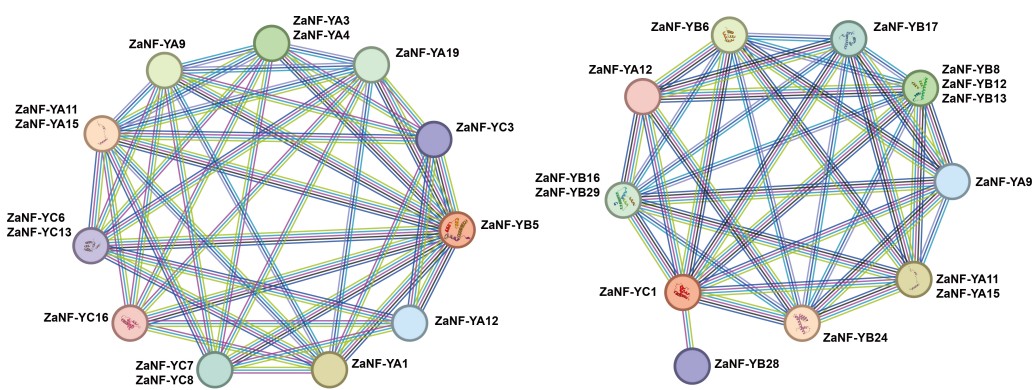

**Figure 9** Protein–protein association network analysis of ZaNF-Y proteins visualized by STRING.

alfalfa, and poplar (*Li et al., 2019*; *Jiang et al., 2023*; *Li et al., 2023*; *An et al., 2022*; *Liu et al., 2021*). Despite their importance, NF-Y family members in *Z. armatum* have remained unexplored. Using the diploid genome of *Z. armatum*, we identified 67 *ZaNF-Y* genes: 20 *ZaNF-YA*, 30 *ZaNF-YB*, and 17 *ZaNF-YC* members (File S9, Fig. 1). This represents a significant expansion compared to 36 *NF-Y* genes in *Arabidopsis* 34 in rice, and 24 in pummelo. Additionally, *ZaNF-Y* transcription factors exhibit conserved domain and motif compositions consistent with their corresponding subunits, demonstrating their evolutionary conservatism (Fig. 5).

Gene duplication drives evolutionary diversification and is a key mechanism for expanding gene families (*Lawton-Rauh, 2003*). In this study, we identified 16 pairs of segmentally duplicated genes and two tandem duplicates in the *ZaNF-Y* family (Fig. 3). All segmentally duplicated gene pairs were found to belong to the same phylogenetic groups as expected, which indicated that the expansion of the ZaNF-Y family was driven by segmental duplications. These duplications have shaped the functional diversity of *ZaNF-Y* genes, with some members acquiring new functions or losing existing ones (*Xu et al., 2020*). For example, in the segmental duplication genes, *ZaNF-YC11* shows significantly low expression in female flowers, whereas *ZaNF-YC16* is upregulated in both female flowers, leaves, and young fruits (Fig. 8). In another pair of segmental duplications, the expression levels of *ZaNF-YB8* and *ZaNF-YB21* in female flowers show an opposite trend (Fig. 8). Tandem duplicates *ZaNF-YC12* and *ZaNF-YC13* show distinct expression patterns: *ZaNF-YC12* is minimally expressed in female flowers, while *ZaNF-YC13* is highly expressed in young fruits, suggesting functional adaptation to reproductive roles (Fig. 8). Some duplicated gene pairs indicated consistent expression patterns, while others did not. For instance, *ZaNF-YC1* is highly expressed in young fruits and female flowers but not in stems, whereas *ZaNF-YC9* depicts no significant expression difference across tissues. *ZaNF-YA1* and *ZaNF-YA8* are significantly downregulated in female flowers, but *ZaNF-YA1* is also downregulated in leaves and young fruits. In the female flowers of *Z. armatum*, the nucellus cells initiate embryo differentiation, whereas in the young fruit, the nucellar embryo has already begun to develop. The initiation of embryo differentiation and

development is typically regulated by different genes. Therefore, these gene duplications may have allowed the *ZaNF-Y* genes to play different regulatory roles at various stages of nucellar embryo development.

NF-Y transcription factors are crucial in the formation of somatic embryos. Ectopic expression of the *LEC1* gene in transgenic plants induces somatic embryo-like structures (*Harada, 2001*). *GhL1L1*, a LEC1-like gene, regulates somatic embryogenesis in cotton by affecting auxin distribution (*Xu et al., 2019*). The heterologous expression of *OsNF-YB7* in the *Arabidopsis lec1* mutant can complement the lec1 deficiency (*Niu et al., 2021*). Phylogenetic analysis depicts that *ZaNF-YB5* is homologous to *AtLEC1* and *OsNF-YB7* (Fig. 1). However, *ZaNF-YB5*'s low expression in female flowers suggests a divergent regulatory role compared to its homologs in *Arabidopsis*, such as *AtLEC1*, which is essential for somatic embryogenesis (Fig. 8). This may result from functional divergence in different species. Interestingly, *ZaNF-YC1* is highly expressed in female flowers and young fruits, indicating it may also play a positive regulatory role in nucellar embryo formation (Fig. 8).

In the *Arabidopsis lec1* mutant, the expression level of the *BBM* gene is significantly down-regulated, while *BBM* expression in rice ovules induces the formation of apomixis (*Pelletier et al., 2017*; *Khanday et al., 2019*). Therefore, we speculate that ZaNF-YB5 may regulate the process of asexual reproduction in *Z. armatum* by affecting the expression of *BBM*. In the *Z. bungeanum*, the MADS transcription factor *AGL11* has been reported to be associated with apomixis (*Fei et al., 2021*). According to the cis-element analysis results, we also found that the promoter regions of many *ZaNF-Y* genes contain binding sites for MADS transcription factors, including *ZaNF-YB5* and *ZaNF-YC1*. Therefore, we also speculate that *AGL11* may affect the apomixis in *Z. armatum* by regulating the expression of *ZaNF-YB5* and *ZaNF-YC1*. We will further validate the function of these candidate genes in nucellar embryo formation by overexpressing them in other model plants (*e.g.*, *Arabidopsis*) or mutating these genes in *Z. armatum* using CRISPR/Cas9 technology.

Studies have found that hormones are capable of inducing the formation of embryos from plant somatic cells (*Long et al., 2018*). ABA has been proven to induce cell differentiation during the somatic embryogenesis process in *Cunninghamia lanceolate* (*Zhou et al., 2017*). In *Arabidopsis*, an increase in JA content inhibits *MYC2* and activates the expression of *JAZ1*, thereby initiating the formation of somatic embryos (*Mira et al., 2016*). Similarly, the promoter regions of the *ZaNF-Y* genes also contain abundant ABA and JA responsive elements, suggesting that they may play a significant role in nucellar embryo formation (Fig. 6). Polar auxin transport and auxin responses are crucial for specifying embryogenic cell fate (*Jia et al., 2021*). In this study, we also found that some *ZaNF-Y* genes' promoter contain auxin response elements. These *ZaNF-Y* genes may participate in the formation of ovule embryos by responding to auxin signals. Additionally, the promoter regions of the *ZaNF-Y* genes also contain a large number of abiotic stress response elements. This may be related to the important regulatory role of ZaNF-Y genes in abiotic stress.

The function of NF-Y requires the formation of a heterotrimeric complex involving NF-YA/B/C subunits, which exhibit complex and specific interactions (*You et al., 2021*). In *Arabidopsis*, 10 AtNF-YB proteins (AtNF-YB1 to AtNF-YB10) and seven AtNF-YC proteins (AtNF-YC1 to C4, C6, C9, and C12) are strongly interconnected, with 74%

of theoretical protein-protein interactions detected by yeast two-hybrid experiments (*Hackenberg et al., 2012*). In this system, AtNF-YBs and AtNF-YCs rarely interacted with AtNF-YAs, indicating that the AtNF-YB/C heterodimer must form to bind AtNF-YA subunits (*Hackenberg et al., 2012*). Similar interactions have been observed in rice between OsHAP3A and two OsHAP2s, as well as six OsHAP5s (*Thirumurugan et al., 2008*). In *Arabidopsis*, AtNF-YA can only form heterotrimeric complexes with dimers containing the HFD subunit and cannot bind to individual AtNF-YA or AtNF-YB subunits (*Hackenberg et al., 2012*). However, studies indicate that OsNF-YA8 can interact with OsNF-YB9 through yeast-two-hybrid assays. In this study, we found that ZaNF-YA11, ZaNF-YA12, and ZaNF-YA15 have potential interactions with many ZaNF-YB subfamily proteins (Fig. 9). NF-YA, NF-YB, and NF-YC subfamily proteins form heterotrimeric complexes, which then interact with other NF-YA proteins or regulatory factors to regulate downstream target gene expression (*Gnesutta et al., 2017*; *Nardone, Chaves-Sanjuan & Nardini, 2017*; *Huang et al., 2015*). Specifically, ZaNF-YB5 may interact with ZaNF-YA9 and ZaNF-YA12, while ZaNF-YA9 and ZaNF-YA12 may also interact with ZaNF-YC1, suggesting the formation of heterotrimeric complexes involving ZaNF-YA9, ZaNF-YA12, ZaNF-YB5, and ZaNF-YC1. In *Arabidopsis*, LEC1 can interact with AtNF-YC2 (*Hackenberg et al., 2012*). Therefore, ZaNF-YB5 (the homolog of LEC1) and ZaNF-YC1 (the homolog of AtNF-YC2) may also be able to form a complex. Moreover, AtNF-YC2, AtNF-YA2, and AtNF-YA4 have the same expression pattern, suggesting that they may function together. Similarly, ZaNF-YA9, ZaNF-YA12, ZaNF-YB5, and ZaNF-YC1 may also participate in nucellar embryo formation by forming complexes in *Z. armatum*, We will further validate these interactions through experiments (such as yeast two- and three-hybrid analysis, *etc.*).

## CONCLUSIONS

In this study, we identified 67 *ZaNF-Y* transcription factors from the diploid *Z. armatum* genome, including 20 *ZaNF-YA*, 30 *ZaNF-YB*, and 17 *ZaNF-YC* genes. We analyzed their basic characteristics, chromosomal localization, and gene duplication events. These *ZaNF-Y* genes demonstrated functional similarity and specificity based on conserved structural domains and motif analysis. Transcription factor analysis suggested potential regulatory mechanisms of *ZaNF-Y* genes at the transcriptional level. Differential expression analysis across various tissues indicated the regulatory roles of specific *ZaNF-Y* members (*ZaNF-YB5* and *ZaNF-YC1*) during nucellar embryo formation. Furthermore, protein association network analysis provided insights into the composition of ZaNF-Y complexes. Overall, the identification of *ZaNF-Y* genes associated with nucellar embryo formation provides a basis for harnessing apomixis in hybrid crop production.

## ACKNOWLEDGEMENTS

We thank Home for Researchers for language editing services.

### Funding

This research was supported by Natural Science Foundation of Sichuan Province (No. 2022NSFSC0089), Key Research Projects of Science and Technology Bureau of Leshan Town (No. 22ZDYJ0082), Scientific research project of Leshan Normal University (No. RC2023005), and Scientific research and cultivation project of Leshan Normal University (No. KYPY2024-0011). The funders had no role in study design, data collection and analysis, decision to publish, or preparation of the manuscript.

### Grant Disclosures

The following grant information was disclosed by the authors:
Natural Science Foundation of Sichuan Province: No. 2022NSFSC0089.
Key Research Projects of Science and Technology Bureau of Leshan Town:  No. 22ZDYJ0082.
Scientific research project of Leshan Normal University: No. RC2023005.
Scientific research and cultivation project of Leshan Normal University: No. KYPY2024-0011.

### Competing Interests

The authors declare there are no competing interests.

### Author Contributions

- Xianzhe Zheng conceived and designed the experiments, performed the experiments, analyzed the data, prepared figures and/or tables, and approved the final draft.
- Yanling Duan performed the experiments, prepared figures and/or tables, and approved the final draft.
- Huifang Zheng performed the experiments, prepared figures and/or tables, and approved the final draft.
- Hao Tang performed the experiments, prepared figures and/or tables, and approved the final draft.
- Liumeng Zheng performed the experiments, prepared figures and/or tables, and approved the final draft.
- Xiaobo Yu conceived and designed the experiments, authored or reviewed drafts of the article, and approved the final draft.

### Data Availability

The raw data are available in the Supplemental Files.

### Supplemental Information

Supplemental information for this article can be found online at http://dx.doi.org/10.7717/peerj.19142#supplemental-information.

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
