# Peer review of "Genome-wide identification and characterization of the NF-Y proteins in Zanthoxylum armatum"

_PeerJ, doi:10.7717/peerj.19142_

## Round 0.1 · original submission · Minor Revisions

Please go through the comments of all reviewers and provide a point wise response to reviewers along with revising manuscript accordingly before its resubmission.

Reviewer 1 ·

Basic reporting

This manuscript by Zheng et. al., investigates the role of Nuclear Factor Y (NF-Y) transcription factors in somatic embryo formation of Zanthoxylum armatum, a plant known for its economic and medicinal value.

Experimental design

The methodology is well-detailed, particularly for identifying and analyzing ZaNF-Y genes. The inclusion of specific tools (e.g., HMMER for gene prediction, WoLF PSORT for localization) is good practice.
The Materials and Methods section could be improved by consolidating redundant information and maintaining a clear focus on each procedure. For example, breaking down the RNA extraction and qRT-PCR processes into separate, labeled subsections would improve organization.
The phylogenetic analysis and promoter cis-element analysis sections are clearly explained but could include details on parameter choices, such as why certain phylogenetic methods or thresholds were used.
Statistical analysis descriptions are adequate, but the rationale for using three biological and technical replicates could be discussed.

Validity of the findings

No comment

Additional comments

Strengths of the manuscript:
• The manuscript focuses on a specific question: How do NF-Y transcription factors contribute to somatic embryogenesis in Z. armatum?
• The authors employ various techniques to analyze NF-Y genes in Z. armatum, including identification, phylogenetic analysis, chromosomal location, conserved domains, cis-element analysis, and expression profiling.
• The study explores potential interactions between NF-Y subfamily members using protein association network analysis.
• Understanding NF-Y involvement in apomixis could contribute to seedless fruit production and breeding strategies.
Apart from these strengths here are the few weakness which if addressed would improve the manuscript significantly:
• While the study identifies potential roles for specific NF-Y genes, functional validation through techniques like overexpression or knockout experiments is missing.
• The manuscript heavily relies on in silico analysis. Including in vitro experiments to support the predicted interactions and expression patterns could strengthen the findings.
• Although the study explores expression in flowers and fruits, additional tissue-specific expression data could strengthen the connection to apomixis.
• The role of NF-Y in plant development is well established. The manuscript would benefit from a clearer discussion on the novelty of the findings specific to Z. armatum apomixis.
Also, following points can also be considered:
• Include functional validation experiments (e.g., overexpression or knockout) to confirm the roles of key NF-Y genes in somatic embryogenesis.
• Consider in vitro experiments to support the predicted protein-protein interactions.
• Analyze tissue-specific expression of NF-Y genes beyond flowers and fruits to further link them to apomixis.
• Enhance the discussion on the novelty of the findings by emphasizing the specific contribution to understanding apomixis in Z. armatum compared to existing knowledge of NF-Y in plant development.

Reviewer 2 ·

Basic reporting

Comment:
Clarity and Language:
The manuscript is well-written, but there are some areas where the language could be refined for better readability.
Comment:
1- Consider simplifying overly complex phrases to enhance readability. Improve the flow and clarity of technical sections to make the content more accessible. Condense repetitive elements while retaining the scientific rigor to ensure brevity without compromising the quality or precision of the information.

For example:
Some sentences in the Introduction, method, results and discussion as well as in abstract sections are overly complex and could be simplified to ensure accessibility for a broader audience.

Examples of areas in the manuscript where the language could be improved for clarity and accessibility, along with suggested revisions:

1. Introduction Section
Original Text: Lines 32–39
“Zanthoxylum armatum DC., an aromatic crop of the Rutaceae family, is widely used in traditional Chinese medicine and cuisine, holding significant economic value (Hui et al., 2020). Several Zanthoxylum species, including Z. armatum, exhibit sporophytic apomixis, forming somatic embryos from the nucellus, similar to Citrus plants (Hojsgaard & Pullaiah, 2022). Reports on apomixis in Zanthoxylum have identified genes such as Zardc07021 in Z. armatum, homologous to the CitRWP gene associated with apomixis in Citrus (Wang et al., 2017).”
Suggested Revision:
“Zanthoxylum armatum DC., an aromatic plant of the Rutaceae family, is economically important for its applications in traditional Chinese medicine and cuisine (Hui et al., 2020). Several species in the Zanthoxylum genus, including Z. armatum, exhibit sporophytic apomixis, where somatic embryos develop from nucellar cells, a process similar to that in Citrus plants (Hojsgaard & Pullaiah, 2022). Studies on apomixis in Zanthoxylum have identified genes such as Zardc07021, which is homologous to the CitRWP gene in Citrus, known for its role in apomixis (Wang et al., 2017).

Original Text: Lines 46–55
“Nuclear factor Y (NF-Y), also known as the CCAAT-box binding factor (CBF) or heme-associated protein, is a class of transcription factors present in eukaryotes (Mantovani, 1999; Myers & Holt, 2018). It comprises three subfamilies: NF-YA, NF-YB, and NF-YC (Nardini et al., 2013). Typically, NF-YB and NF-YC proteins form a heterodimer in the cytoplasm and then bind to the NF-YA protein in the nucleus to create a complete NF-Y heterotrimer complex (Gnesutta et al., 2017). This complex regulates downstream target gene expression via the N-terminal of the NF-YA subunit or by binding other regulatory factors (Nardone, Chaves-Sanjuan & Nardini, 2017; Huang et al., 2015). NF-Y genes significantly influence plant growth, development, stress responses, and hormone signaling pathways (Niu et al., 2021; Wang et al., 2021b; Ke et al., 2022)”
Suggested Revision:
“Nuclear factor Y (NF-Y), also called the CCAAT-box binding factor (CBF) or heme-associated protein, is a family of transcription factors conserved across eukaryotes (Mantovani, 1999; Myers & Holt, 2018). The family consists of three subunits: NF-YA, NF-YB, and NF-YC (Nardini et al., 2013). Typically, NF-YB and NF-YC form a heterodimer in the cytoplasm, which then binds to NF-YA in the nucleus to assemble a functional NF-Y heterotrimer complex (Gnesutta et al., 2017). This complex regulates downstream gene expression either through the NF-YA subunit’s N-terminal or by interacting with other regulatory proteins (Nardone et al., 2017; Huang et al., 2015). NF-Y transcription factors play vital roles in plant growth, development, stress responses, and hormone signaling (Niu et al., 2021; Wang et al., 2021b; Ke et al., 2022)”

Original Text: Lines 62–69
“In recent decades, the role of NF-Y family members in embryo development has been extensively studied. LEAFY COTYLEDON 1 (LEC1, AtNF-YB9) is a transcription factor crucial for embryo maturation, with a distinctive expression pattern in embryos (Lotan et al., 1998; Lee et al., 2003). Ectopic expression of the LEC1 gene in transgenic plants induces the formation of somatic embryo-like structures (Harada, 2001). In rice, the LEC1 homolog OsNF-YB7 is primarily expressed in the embryo, and its knockout leads to abnormal embryo development (Niu et al., 2021). Other family members, such as LEC1-like (LIL, NF-YB6), also play significant roles in embryo development (Kwong et al., 2003). In Arabidopsis, NF-YA3 and NF-YA8 exhibit functional redundancy and are essential for embryonic development (Fornari et al., 2013).”
Suggested Revision:
“In recent decades, NF-Y family members have been extensively studied for their roles in embryo development. LEAFY COTYLEDON 1 (LEC1, AtNF-YB9), a key transcription factor, is essential for embryo maturation and exhibits a distinct expression pattern in embryonic tissues (Lotan et al., 1998; Lee et al., 2003). Overexpression of LEC1 in transgenic plants induces somatic embryo-like structures (Harada, 2001). Similarly, in rice, the LEC1 homolog OsNF-YB7 is predominantly expressed in embryos, and its disruption results in abnormal embryo formation (Niu et al., 2021). Other related proteins, like LEC1-like (LIL, NF-YB6), also play critical roles in embryo development (Kwong et al., 2003). In Arabidopsis, NF-YA3 and NF-YA8 are functionally redundant but collectively necessary for embryogenesis (Fornari et al., 2013).”

2.0 Methods:
Original Text: qRT-PCR Analysis.
Suggested Revision: Why do you use another endogenous control, GAPDH, instead of beta actin? Please rewrite for more precision.

3. Discussion Section
Original Text: Lines 243–253
“Extensive research has explored NF-Y transcription factors and their roles in various biological processes (Li et al., 2019). In species such as cabbage, melon, alfalfa, and poplar, NF-Y family members have been identified and analyzed for their functions in flowering, abiotic stress response, growth, and development (Jiang et al., 2023; Li et al., 2023; An et al., 2022; Liu et al., 2021). However, NF-Y family members in Z. armatum have not been reported. This study utilized the published diploid genome of Z. armatum to identify and analyze ZaNF-Y transcription factors. We identified 67 ZaNF-Y genes, including 20 ZaNF-YA, 30 ZaNF-YB, and 17 ZaNF-YC genes (File S9, Figure 1). Compared to 36 Arabidopsis AtNF-Y genes, 34 rice OsNF-Y genes, and 24 pummelo CgNF-Y genes, the ZaNF-Y genes in Z. armatum have significantly expanded.”
Suggested Revision:
“NF-Y transcription factors play vital roles in biological processes such as flowering, stress response, growth, and development, as shown in species like cabbage, melon, alfalfa, and poplar (Li et al., 2019; Jiang et al., 2023; Li et al., 2023; An et al., 2022; Liu et al., 2021). Despite their importance, NF-Y family members in Zanthoxylum armatum have remained unexplored. Using the diploid genome of Z. armatum, we identified 67 ZaNF-Y genes: 20 ZaNF-YA, 30 ZaNF-YB, and 17 ZaNF-YC members (File S9, Figure 1). This represents a significant expansion compared to 36 NF-Y genes in Arabidopsis, 34 in rice, and 24 in pummelo.

Original Text: Lines 254–262
“Evolutionary diversification is often driven by gene duplication, which plays a crucial role in expanding gene families (Lawton-Rauh, 2003). In this study, we identified 16 pairs of segmental duplications and two tandemly duplicated genes, suggesting that segmental duplication significantly influenced the evolutionary trajectory of the ZaNF-Y family (Figure 3). Duplicated gene pairs commonly become functionally diverse, with some members acquiring new functions or losing original ones (Xu et al., 2020). For instance, the tandemly duplicated genes ZaNF-YC12 and ZaNF-YC13 have undergone functional divergence: ZaNF-YC12 is lowly expressed in female flowers, while ZaNF-YC13 is highly expressed in young fruits (Figure 8).
Suggested Revision:
“Gene duplication drives evolutionary diversification and is a key mechanism for expanding gene families (Lawton-Rauh, 2003). In this study, we identified 16 pairs of segmentally duplicated genes and two tandem duplicates in the ZaNF-Y family (Figure 3). These duplications have shaped the functional diversity of ZaNF-Y genes, with some members acquiring new functions or losing existing ones (Xu et al., 2020). For example, tandem duplicates ZaNF-YC12 and ZaNF-YC13 show distinct expression patterns: ZaNF-YC12 is minimally expressed in female flowers, while ZaNF-YC13 is highly expressed in young fruits, suggesting functional adaptation to reproductive roles (Figure 8).

4. Figure Legends: Figure legends should be more specifically addressed in the context of researcher in the same field, especially if the target audience consists of non-specialists.

Experimental design

Comments:
Phylogenetic Analysis:
Observation: While the method mentions the use of maximum likelihood estimation, there is no explanation of how the phylogenetic reliability was assessed.
Specific Suggestions:
Include the bootstrap threshold used to evaluate phylogenetic tree reliability (e.g., “Only nodes with bootstrap values above 70% were considered reliable”).
Explain the criteria for selecting homologous NF-Y genes from Arabidopsis, rice, and Citrus (e.g., based on sequence similarity or functional conservation).
Promoter Analysis:
Observation: The manuscript briefly mentions the use of PlantCARE and PlantPAN for cis-element analysis but does not explain why these tools were chosen.
Specific Suggestions:
Justify the choice of these tools by discussing their capabilities (e.g., PlantCARE’s database of hormone-responsive cis-elements, PlantPAN’s focus on transcriptional regulation in plants).
Highlight the significance of identifying hormone- and stress-responsive cis-elements in ZaNF-Y gene promoters in the context of nucellar embryo formation.

qRT-PCR Validation: Suggested Revision: Why do you use another endogenous control, GAPDH, instead of beta actin? Please rewrite for more precision.

Validity of the findings

Comments:
Gene Duplication Analysis:
Observation: While the manuscript identifies 16 segmental duplications and 2 tandem duplications, it does not provide a detailed evolutionary interpretation.
Specific Suggestions:
Discuss how segmental duplications may have contributed to the expansion of the ZaNF-Y family and its functional diversity.

Functional Predictions:
Observation: ZaNF-YC1 is identified as a potential positive regulator of nucellar embryo formation, but the lack of functional validation is a limitation.

Specific Suggestions:
Acknowledge this limitation explicitly in the Discussion and propose future directions, such as overexpression or CRISPR-based knockout experiments to confirm ZaNF-YC1’s role.

Expression Data:
Observation: The tissue-specific expression of ZaNF-Y genes is presented, but the implications of this data are not fully explored.
Specific Suggestions:
Elaborate on the biological significance of ZaNF-YB5’s downregulation in female flowers. Example: “ZaNF-YB5’s low expression in female flowers suggests a divergent regulatory role compared to its homologs in Arabidopsis, such as AtLEC1, which is essential for somatic embryogenesis.”

Additional comments

Integration of Findings:
• Connect the results more explicitly to potential applications in agriculture or biotechnology. For example:
“The identification of ZaNF-Y genes associated with nucellar embryo formation provides a basis for harnessing apomixis in hybrid crop production.”


• Define all abbreviations at their first use, especially for technical terms and software (e.g., MCScanX, PlantPAN).
• Standardize gene nomenclature throughout the text (e.g., always italicize ZaNF-Y gene names).

The manuscript is scientifically robust and presents valuable insights into NF-Y genes in Zanthoxylum armatum. Addressing the comments above will enhance its clarity, depth, and impact.

Reviewer 3 ·

Basic reporting

This manuscript reporting on the 'Genome-Wide Identification and Characterization of the NF-Y proteins in Zanthoxylum armatum' has reported on the NF-Y proteins and their characterization. It is written very well, and experiments are conducted suitably to draw conclusions.
I have following observations for further improvement.

1. The CIS acting elements: authors reported 21 elements. Can they be classified into the five categories, which normally cover growth and development, hormone response, light response, stress response, and other cis-type components.
2. Authors studied Tissue-specific Expression of ZaNF-Y genes looking at expression in different tissues/organs. They have mentioned 'somatic embryogenesis' (somatic embryogenesis through tissue expression variations). Can they check the usage the term once again. Its normally the in vitro embryogenesis through callus/direct in vitro which can be termed so. It should be nucellar embryos.
3.Few recent reviews can be checked for updates. For ex. Kavi Kishor, et al. Nuclear Factor-Y (NF-Y): Developmental and Stress-Responsive Roles in the Plant Lineage. J Plant Growth Regul 42, 2711–2735 (2023). Zhang et al.. Crucial Abiotic Stress Regulatory Network of NF-Y Transcription Factor in Plants. Int J Mol Sci. 2023 Feb 23;24(5):4426. doi: 10.3390/ijms24054426.
4.

Experimental design

Experimental design is appropriate for the study.

Validity of the findings

Very good, and useful data on the NF-Y genes, their characterization and expression

Additional comments

This manuscript reporting on the 'Genome-Wide Identification and Characterization of the NF-Y proteins in Zanthoxylum armatum' has reported on the NF-Y proteins and their characterization. It is written very well, and experiments are conducted suitably to draw conclusions.

Reviewer 4 ·

Basic reporting

no commen

Experimental design

no commen

Validity of the findings

no comment

Additional comments

Somatic embryogenesis is an important biological process in many plant species, especially in economically important plants like Zanthoxylum armatum. Studying the effects of NF-Y transcription factors on this process not only provides insight into their roles in development, but also provides a theoretical basis for plant biotechnology and breeding research. Therefore, this study has important biological value and application potential.

I have few questions regarding the manuscript:

Although the article presents genetic studies of Z. armatum in gametophyte-free reproduction, a detailed discussion of the uniqueness of Z. armatum is lacking when comparing it to other species. It is recommended that the authors further elaborate on the similarities and differences between Z. armatum and other plants in the mechanism of gametophyte-free reproduction. (lines 36-38)

The word "heterogeneous" seems out of place. "Heterologous" is likely the intended term. (lines 38)

The manuscript references multiple studies across different plant species (soybean, Arabidopsis, Populus), but a more explicit comparison of how NF-Y function differs or is conserved among these species would strengthen the manuscript. Discussing potential species-specific differences in NF-Y-mediated stress responses or growth regulation could provide valuable insights into the broader role of NF-Y in plant biology. (lines 57-61)

The manuscript mentions the lack of understanding regarding the role of NF-Y transcription factors in somatic embryogenesis in Z. armatum. It would strengthen the manuscript to include a brief discussion on potential approaches or methods that could be used to investigate NF-Y function in this species, such as transcriptomic analysis or CRISPR-based knockouts, to fill this knowledge gap. (lines 65-70)

The use of HMMER software with a cut-off value of 0.01 for predicting ZaNF-Y proteins is mentioned, but it would be helpful to explain why this specific cut-off value was chosen. Providing some context on how this cut-off was determined (e.g., based on previous studies or optimization for Z. armatum) would add clarity and strengthen the methodological rigor. (lines 88)

(https://doi.org/10.6084/m9.figshare.14400884.v1), (http://pfam-legacy.xfam.org/) may be in the form of a reference. (lines 86,87,96)

Providing the parameters used for the phylogenetic tree construction (e.g., substitution model, gap penalties) would increase the reproducibility and transparency of the analysis. (lines 108)

The manuscript mentions the use of TBtools-II and MCscanX for visualizing chromosomal locations, segmental duplications, and synteny analysis, but it would be beneficial to specify the parameters or settings used in these tools (e.g., minimum similarity threshold for duplications, synteny block size). This would help readers assess the robustness and reproducibility of the analyses. (lines 113-114)

What statistical tests are used to analyze qRT-PCR data? (lines 146)

While the manuscript identifies and classifies the ZaNF-Y genes, further functional characterization or potential roles of these genes in Z. armatum would strengthen the study. For example, discussing preliminary insights into their involvement in stress response, development, or other biological processes could provide a more complete understanding of the functional significance of these genes. (lines 250-254)

The manuscript presents a variety of expression patterns for duplicated gene pairs (e.g., ZaNF-YC1, ZaNF-YC9), but the biological significance of these patterns could be more thoroughly discussed. It would be helpful to hypothesize how the differential or non-differential expression of these genes might relate to specific developmental or environmental processes in Z. armatum, providing more context to the findings. (lines 264-265)

The manuscript proposes that ZaNF-YC1 may play a role in embryo development due to its high expression in female flowers and young fruits. To strengthen this conclusion, the authors could discuss potential mechanisms by which ZaNF-YC1 may influence embryo development (e.g., through hormonal regulation or interaction with other key transcription factors) and whether this expression pattern aligns with known regulators of somatic embryogenesis in other species. (lines 277-278)

Functional Implications of Interactions: The manuscript suggests that ZaNF-YA9, ZaNF-YA12, ZaNF-YB5, and ZaNF-YC1 may form heterotrimeric complexes that regulate embryo development. To strengthen this hypothesis, the authors could discuss the functional roles of these complexes in more detail, including how they might influence key pathways in embryo development (e.g., hormonal regulation, gene expression) or whether these interactions are conserved across other species. (lines 296-298)

"Besides, protein association network analysis provided insights into the composition of ZaNF-Y complexes." Suggestion: "Besides" is somewhat informal. Consider using "Additionally" or "Furthermore" for a more formal tone. (lines 310)

---

## Round 0.2 · Minor Revisions

We apologize for the premature Accept decision you received.

Upon further inspection by a Section Editor, the manuscript requires minor revisions before final acceptance. Please address the following:
1. Provide a more detailed justification for the selection of your analytical tools and methods
2. Expand the description of these tools' capabilities and limitations
3. Enhance the methodology section to improve clarity for readers in the field

We look forward to your revised manuscript.

Regards,

Reviewer 2 ·

Basic reporting

Thank you for addressing my previous comments. I appreciate the effort made in revising the manuscript

Experimental design

Thank you for addressing my previous comments. I appreciate the effort made in revising the manuscript

Validity of the findings

Thank you for addressing my previous comments. I appreciate the effort made in revising the manuscript. However, some points still require further clarification, especially if you aim to enhance readability for other scholars in this field. In particular, please justify the choice of these tools by discussing their capabilities in more detail. That being said, I defer to the editor’s judgment on this matter.

Reviewer 3 ·

Basic reporting

The revised version is quite improved in different aspects as suggested by the reviewers

Experimental design

The revised version is quite improved in different aspects as suggested by the reviewers

Validity of the findings

Good

---

## Round 0.3 · accepted · Accept

Authors have addressed all of the reviewers' comments, and the manuscript is ready for publication.